# Experimental Studies on Fabricating Lenslet Array with Slow Tool Servo

**DOI:** 10.3390/mi13101564

**Published:** 2022-09-21

**Authors:** Wenjun Kang, Masafumi Seigo, Huapan Xiao, Daodang Wang, Rongguang Liang

**Affiliations:** 1Wyant College of Optical Science, The University of Arizona, Tucson, AZ 85719, USA; 2State Key Laboratory of Ultra-Precision Machining Technology, Department of Industrial and System Engineering, The Hong Kong Polytechnic University, Hung Hom, Kowloon, Hong Kong, China

**Keywords:** diamond turning, slow tool servo, lenslet array, position following error, optics fabrication

## Abstract

On the demand of low-cost, lightweight, miniaturized, and integrated optical systems, precision lenslet arrays are widely used. Diamond turning is often used to fabricate lenslet arrays directly or molds that are used to mold lenslet arrays. In this paper, mainly by real-time monitoring position following error for slow tool servo, different fabrication parameters are quantitatively studied and optimized for actual fabrication, then by actual fabrication validation, uniform and high-fidelity surface topography across the actual whole lenslet array is achieved. The evaluated fabrication parameters include sampling strategy, inverse time feed, arc-length, etc. The study provides a quick, effective, and detailed reference for both convex and concave lenslet array cutting parameter selection. At the end, a smooth zonal machining strategy toolpath is demonstrated for fabricating concave lenslet arrays.

## 1. Introduction

Lenslet arrays are commonly used in imaging systems to miniaturize the system, such as using lenslet arrays to obtain uniform illumination in illumination systems [1] or to sample the wavefront in a wavefront sensing system [2]. Single-step photolithography assisted by the chemical wet etching method has been reported to fabricate multifocal micro-lens arrays [3]. CO_2_ laser-assisted fabrication of micro-lens arrays and characterization of their beam-shaping property has also been researched [4]. Micro-lens arrays based on all-silicon metasurfaces by photolithography and etching for long-wavelength infrared application has also been reported [5]. Alternatively, through computer controlled multi-axis synchronized ultra-precise motion, single point diamond turning can also fabricate lenslet arrays, since it can achieve nanometer-level surface roughness and micrometer-level form accuracy. Compared to fabricating axial symmetrical surfaces, fabrication of lenslet arrays is more difficult because the surface is not continuous. The sharp transition edge between two lenslets makes the fabrication of a lenslet array even more challenging. The diamond turning process is mainly performed with slow tool servo (STS). To achieve larger bandwidth of tool stroke frequency, fast tool servo (FTS) has been developed to fabricate the surface with rapid variation features [6,7]. The drawback is that FTS suffers from limited travel distance, which means that the peak to valley of the surface it can generate is limited [6]. Another limitation is that not every machine has FTS, as it is expensive. Therefore, it is still necessary and worth further study and characterization of the conventional STS configuration to rapidly prototype complex geometry like a lenslet array and achieve a more precise level.

There are varieties of factors that can affect the final fabricated surface finish, such as cutting parameters, tool paths, temperature, cutting fluids, external vibration, and the vibration between the diamond tool and workpiece [8,9,10,11,12,13,14,15]. Poor surface finish is notoriously associated with random light scattering and undesired stray light [16,17]. A lot of work has been done to improve the surface finish, including machining accuracy improvement [18] and tool path and tool compensation studies [19,20]. In particular, much effort and research has been poured into lenslet array fabrication. Yi reported around 1 µm deviation among scanned curves of three separate lenslet profiles and 34.5 nm Ra surface roughness [21]. Chen utilized the sliding method to cut the lens array with STS [22], in which the machine-axis-orientation dependent form error and the increase in the error at the edge region of the single lens was observed. Scheiding foresaw the edge issue and incorporated voice coil FTS and cubic spline interpolation to improve lenslet surface form error, reported to have 31.65 nm RMS in the center region and 39.56 nm RMS in the outer region of the lenslet array [7]. By introducing an additional blend zone through blending and extrapolation polynomials in the single lenslet edge region, Neil reported improved surface quality of the lenslet edge region in the fabrication process [23]. A single crystal silicon IR-application lenslet array with the STS process encountering brittle fracture and associated with poor surface topography can also be greatly optimized [24]. By inserting additional points at the edge of lenslet, Liu showed improved lenslet topographical results for an FTS system [25]. To avoid unwanted light transmitting through the gap space of low-fill-factor lenslet arrays, a film-deposited gap-masked lens array on a curved substrate has been studied as well [26]. Similarly, a low-fill-factor aspheric lenslet array fabricated with STS was reported by Mishra, with 233 nm form accuracy and 14.2 nm (Ra) surface roughness [27].

To further improve fabricated lenslet array surface topography, this paper, by monitoring position following error, quantitatively studies the impact of sampling strategy, inverse time feed, sampling arc length, fabrication efficiency, and convex and concave lenslet configuration difference. Furthermore, using position following error mitigation methodology as guidance, actual lenslet arrays are fabricated and the improvement in surface topography at the outer region and uniformity across the whole array are verified. At the end, a smooth zonal machining toolpath methodology is introduced, and excellent surface form accuracy for both center and outer region concave lenslets is demonstrated.

## 2. Experiment Setup

Here, the study’s geometrical feature is a lenslet array. It comes with two configurations: convex and concave, as shown in Figure 1a,b. These arrays have a high-fill-factor, making the transition between the adjacent lenslets very sharp and posing huge challenges in precision motion control in the machining process.

To generate the tool path for a designed lenslet array surface, considering the diamond turning conventional spiral tool path, we first generated a spiral curve on a flat surface, and then projected the spiral curve to the designed lenslet array surface. After that, taking the tool radius and rake angle into account, the tool path was generated by implementing cylindrical-coordinate based tool compensation [19]. In this process, the two adjacent points in the tool path can be sampled with different strategies. By maintaining constant angular speed, the constant-angle sampling strategy takes full advantage of the accurate angular positioning property of the air bearing spindle. However, it has obvious disadvantage that movement between two adjacent points gets increasingly more challenging within a fixed response time, since the sampled arc length increases as the tool moves away from the spindle center. The fixed response time is defined as the inverse time feed in G-code. The inverse time feed is typically used to program simultaneous coordinated linear and rotary motion. To overcome the limitation, a constant-arc sampling strategy was developed [28]. Although sparse point distribution in the outer region issue has been greatly improved, the center region becomes problematic with insufficient sampling. Naturally, combining constant-angle and constant-arc sampling strategies can balance their limitations.

### 2.1. Lenslet Array Surface Quality Deterioration at Outer Region

To study how well the lenslet can be fabricated, Nanotech 350FG was utilized to fabricate a 14 mm outer-diameter PMMA convex lenslet array, in which each lenslet had a radius of 6.3 mm and a clear aperture 0.35 mm × 0.35 mm. In the first study, a 0.25° constant-angle sampling strategy was applied, corresponding to 1440 points in one rotation cycle. In addition, an inverse time feed of 0.001 s was used. After the fabrication, the sample was measured with a Zygo NewView 8300 white light interferometer, with the measured result shown in Figure 2. It can be clearly seen that the deterioration of the lenslet surface quality grew gradually from the center to outer region.

To minimize the deterioration and determine the underlying root cause, we introduced PMAC (Programmable Multi-Axis Controller) to capture the real-time motion data, including command position, actual position, their associated acquisition time, etc., so that we could go further into analyzing and understanding the fabrication process. The position following error later discussed in this paper is the difference between the command and actual position.

### 2.2. Position Following Error Tracking

By using PMAC, real-time process data were acquired. Figure 3 showed a set of measurement data at the outer region for a 14 mm outer-diameter lenslet array, as same as before, in which each singlet lens had a radius of 6.3 mm and a clear aperture of 0.35 mm × 0.35 mm. By comparing the coordinates, it was determined that the peaks and valleys shown in Figure 3 correspond to the motion of entering and exiting a lenslet for an STS machine tool when it was going through edges formed by two adjacent lenslets. The position following error indicated the discrepancy between the command and actual motion of the machine tool, which further reflects the deviation between designed surface features and fabricated ones.

Now, knowing the lenslet surface quality deteriorates from center to edge and that the center maintains fidelity to the actual design very well, the whole lenslet array uniformity can be identified if an indicator of outer-region lenslet surface topography is identified. As mentioned previously, the position following error represents local discrepancy very well, which makes it a potential candidate for the indicator. To check it, using the same methodology used in Figure 3, a close scrutiny was performed. As a result, Figure 4a plots the command and actual positions of the machine tool, using Figure 2’s lenslet array fabrication condition: 0.001 s inverse time feed and 0.25° constant-angle. According to Figure 4a, obviously, the tool did not follow the command trajectory precisely. To view the difference quantitatively, Figure 4b shows the position following error, in which a dramatic error is presented at every sharp transition region. In return, this explains the deterioration of the lenslet surface topography in the outer region. The actual position presented an overshoot phenomenon compared to the command position at every peak and valley. This behavior could be due to STS’s large mass and inertia, or the positioning accuracy of the machine itself could have been poor. This needs to be addressed and will be discussed in a later section.

Although it is a small angle for the 0.25° angle cutting condition, it still exceeds the 30 µm arc length for two adjacent points at the edge of the 14 mm outer-diameter lenslet array, which is a large stroke length considering it is a freeform surface with complex local features. This may lead to the surface feature not being sufficiently sampled. In addition, between two adjacent points, the machine performs linear interpolation and a potential desired curve motion is simplified as a straight line, which is called the linearization error, the impact of which has been studied [28]. However, there is another aspect to the actual execution of linear interpolation in the dynamic process—STS dynamic error—that has not drawn much attention. Only recently has it started gaining traction [29]. That means that even with a given straight line motion, with pre-existing motion, whether the STS is capable of following the defined straight-line trajectory correctly is still a question. Through inverse time feed control, this is studied and discussed in Section 2.2.1. In addition, as mentioned in the previous tool path generation process, by shortening the commanded stroke length, it gives more time for STS to respond per unit length. As a result, the constant-arc will overcome the long stroke length issue at the outer-edge area. From the perspective of characterizing linearization error, studies have been carried out by directly measuring the fabricated sample [28], but not from the perspective of position following error, making it not systematic. Furthermore, considering so many other parameters could be involved in the fabrication process, such as tool alignment and wear, chip removal effectiveness, thermal drift, cutting fluids, etc. [8,9,10,11], the final fabricated part’s surface topography is a combination of all sorts of impacts. Therefore, independent motion trajectory study is necessary, and its significance imposed on the surface topography needs to be quantitatively investigated; this is discussed in Section 2.2.2. Naturally, the combined impact of inverse time feed and constant-arc is directly followed in the same section. Then, the whole position following error optimization parameters further guide actual lenslet array fabrication parameter selection and the measurement results of the actual fabricated parts in return serve for validation feedback, all of which is discussed in Section 2.3.

#### 2.2.1. Position Following Error Versus Inverse Time Feed

With the overshoot phenomenon in Figure 4, one question about STS’s capability of correctly following the command trajectory needs to be answered. To do so, during a time span of one second, position following error at 20 random positions across the region of interest in a stationary state was tested and only nanometer and sub-nanometer peak values were recorded, which is 1000 times less than the results in Figure 4b, almost negligible. Then the question turned to address whether it is dynamic motion related. To do so, similar to the cutting condition in Figure 2 and Figure 4, a 0.25° constant-angle was used, but with a difference: The motion control variable—inverse time feed—was extended to 0.004 s. Then, the dynamic motion data were gathered and analyzed for STS, and the result is shown in Figure 5. Clearly, position following error dropped at least 3 times compared to the results in Figure 4. That means that the error was directly related to the motion, which further related to the acceleration of STS, considering STS’s large mass and inertia. With the improvement, this further implies that STS’s actual response frequency range mostly fell out of the initial 0.001 s command frequency range. However, with the extended 0.004 s time, the overlapped region between the command and actual frequency range was increased. As a result, it produced reduced position following error.

To characterize the inverse time feed more carefully, a further study was conducted. All other parameters remained the same, but only one variable—inverse time feed—was tested, and 0.001 s, 0.002 s, 0.003 s, 0.004 s 0.005 s, and 0.006 s were chosen. For each inverse time feed, the position following error for at least one complete spindle rotation cycle was gathered, and then all peak values demonstrated in Figure 5b were averaged as an indicator. In addition, total G-code running consumption time for finishing a whole lenslet array was recorded, and the results are shown in Figure 6. It can be seen from Figure 6 that the position following error indeed dropped exponentially with the increased inverse time feed. By checking the error, at 0.004 s, 0.005 s, and 0.006 s, they resulted in 34.5 nm, 29.2 nm, and 25.3 nm surface following error, respectively, whereas the time consumption corresponded to 17.8 h, 22.25 h, and 26.7 h, respectively. Considering the time consumption and time-dependent variables like thermal drift [30], one may settle with a value between 0.002 s and 0.004 s for actual fabrication so that reasonable accuracy can be produced within a reasonable amount of time. Obviously, through manipulation of the inverse time feed, one can ensure the command motion frequency falls within the inherited STS response frequency range. It should be completely differentiated from linearization error, which is theoretical geometrical simplification, whereas here, it is fully dynamic motion related.

Alternative to directly controlling the STS inverse time feed for each step, varying two adjacent arc lengths is an indirect method to alter the command frequency. Therefore, from a position following error perspective, a systematical study for arc-length impact was conducted.

#### 2.2.2. Position Following Error Verse Arc Length

Obviously, any arbitrary arc length can be picked, but a small arc length means more stroke step numbers for the same total trajectory. Then, in a practical fabrication environment, a tradeoff needs to be carried out with the goal of achieving a reasonable surface quality for task-specific applications within a reasonable amount of time. To quantitatively understand the role that constant-arc plays in the position following error mitigation process, a study was carried out. In the study, all other parameters remained constant, with only one variable—different arc length—being selected, with representative ones across the spectrum: 0.625 µm, 1.25 µm, 2.5 µm, 5 µm, 10 µm, and 15 µm. For each arc length, similar to Figure 6, position following error and total consumption time were recorded, and the results are given in Figure 7. As indicated in Figure 7, when arc length decreased, the position following error dropped dramatically in logarithm format, whereas the total time consumption, on the contrary, increased exponentially. Specifically, the error dropped almost 412.0 nm when arc length went from 15 µm to 1.25 µm, ending up with a time increase of 25 hours. However, after going from a 1.25 µm arc length to a 0.625 µm one, the time increased by 27 h but the error only dropped 39.7 nm. Considering time-dependent variables like thermal drift [30], it may fully cancel out the benefits of shortening the arc length from 1.25 µm to 0.625 µm for STS, so a 1.25 µm or 2.5 µm arc length can be considered to carry out the finishing cut.

Naturally, the combined effect of the STS response time and constant-arc was studied. With the results of the position following error and machining efficiency demonstrated in Figure 6 and Figure 7, a 2.5 µm constant-arc and an inverse time feed of 0.004 s were selected for further testing. The corresponding result is shown in Figure 8. It clearly indicates that the position following error was greatly reduced to a negligible level compared to those in Figure 4 and Figure 5, as well as compared to the magnitude of the designed geometry feature.

So far, Section 2.2.1 and Section 2.2.2 have quantitatively demonstrated, by controlling inverse time feed and changing arc-length, that the position following error in the outer region of lenslet arrays can be mitigated to a negligible level. The results of scaling the value of inverse time feed or arc length to the position following error value can be seen in Figure 6 and Figure 7. We may be able to further use these parameters as a guide and apply them to actual lenslet array fabrication. To find out whether this outer region’s local parameter would translate to lenslet array global surface topography uniformity improvement, we validated it with actual fabricated samples. That means that a few of the typical fabrication parameters studied above were selected to further verify scaling between surface deterioration at the outer region of the array and position following error.

### 2.3. Fabrication Results Validation

In addition to the sample shown in Figure 2, four additional convex lenslet arrays with different sampling strategies and cutting parameters were fabricated: (1) 0.004 s inverse time feed and 0.25° constant-angle sampling, (2) 0.001 s inverse time feed and 2.5 µm constant-arc sampling, (3) 0.004 s inverse time feed and 2.5 µm constant-arc sampling, and (4) 0.004 s inverse time feed and a combination of 0.25° angle sampling for the center region and 2.5 µm constant-arc sampling for the outer region. To address whether the phenomenon observed in Figure 2 is uniquely associated with a convex lenslet, we also fabricated an extra lenslet array with a concave lenslet using 2.5 µm constant-arc sampling and 0.001 s inverse time feed.

To associate the previous tested position following error with actual fabricated lenslet topography, the outer regions of six cases of the above-fabricated lenslet arrays were characterized, and specifically, the fifth-position lenslet, as shown in Figure 2, was chosen for comparison. Figure 9 shows the measured surface shape, and Figure 10 is the corresponding surface details after removing the designed spherical surface. As discussed in Section 2.2.1, increasing the inverse time feed improved the position following error for the constant-angle, and it resonated well with the surface shape results in Figure 9a,b. Thus, increased inverse time feed indeed translates into improved surface quality. Accordingly, Figure 10b shows the reduced magnitude of surface form error compared to Figure 10a.

For the constant-arc, demonstrated in Figure 9c with a 0.001 s inverse time feed and Figure 9d with a 0.004 s inverse time feed, again, longer inverse time feed achieved better results. The surface form error was also reduced dramatically when comparing Figure 10c with Figure 10d. The negligible position following error shown in Figure 8 was directly translated into high-fidelity surface form in Figure 9d and small surface form error in Figure 10d. In addition, comparing Figure 9a,c, significant improvement was observed when switching from constant-angle to constant-arc. Additionally, hybrid sampling and constant-arc sampling were the same for the outer region, which was verified by the identical results shown in Figure 9d,f, as well as in Figure 10d,f.

The surface error associated with the convex or concave configuration can be fully explained from a comparison of surface shape in Figure 9c,e and Figure 10c,e. They present reversed form error and reversed surface shape, which means the surface error was not related to the lenslet’s convex and concave configuration.

To further investigate the uniformity of lenslet array surface topography from the center region to the outer region, for the six cases of the above fabricated lenslet arrays, the Sq (root-mean-square surface roughness) and Sa (arithmetical-mean surface roughness) of five lenslets was plotted in Figure 11 at the positions indicated in Figure 2. It is obvious that the errors increased dramatically from the center to the edge when using the constant-angle sampling strategy, and longer inverse time feed improved the performance to some degree. At the same time, constant-angle showed better results in the center region compared to the constant-arc. On the contrary, in the outer region, the constant -arc showed better results. This also explains why a hybrid strategy can perform well in both the center and outer regions. Therefore, with position following error assisted optimization, surface deterioration mitigation in the outer region of lenslet arrays can be achieved through longer inverse time feed. Accordingly, uniformity across the whole array is shown by a flatter line with a smaller slope. A similar conclusion can be drawn from comparison of the constant-angle-, constant-arc-, and hybrid-sampling strategies. The hybrid one was superior in achieving uniformity. At the outer region of the arrays, compared to the constant-angle, the constant-arc showed better result because of its shorter stroke length between adjacent points, 2.5 µm vs. 30 µm, which also validates that a smaller arc-length induced position following error can be translated into surface deterioration mitigation. In summary, no matter which parameter it is, as long as the position following error is optimized accordingly, it will indeed be translated and scaled into surface deterioration mitigation.

## 3. Zonal Machining Strategy for Concave Lenslet Array

All of the study in Section 2 is based on the condition that the tool path needs to go through the sharp transition imposed by two adjacent lenslets. To fundamentally get rid of this issue, a zonal machining strategy for a concave lenslet array is studied here. Conventionally, G-code is generated either by analytical description or by the fitted polynomial description, such as non-uniform rational B-splines (NURBS) [19,31,32]. To improve the surface topography in fabricating a concave lenslet array, instead of using a conventional one pass approach going through all the geometrical features, including all the sharp transitions, here we utilized the concave lenslet array specific geometry property, treating four lenslets as a unit group and only cutting one lenslet on one cycle, which means that four separate cycles were needed to finish the entire lenslet array. Only one (Figure 12a) of the four lenslets was selected to fit the NURBS, and the rest of the area was interpolated with continuous surface interpolation [31,32], as is shown in Figure 12b. The advantage of doing this is making the machine tool entering and exiting each lenslet smooth; there is no need for STS to adapt to the sharp transition edges. To make a clear comparison with previous results, the position following error was also gathered, but no sharp peaks or edges were captured, as same as the no-peak region shown in Figure 8b. Besides that, the same lenslet array configuration and hybrid sampling strategy with 0.001 s inverse time feed were used to fabricate the lenslet array to validate the results. The measured lenslet surface shape and form error are shown in Figure 13 for the lenslets in the center and outer regions. According to Figure 13, the surface form errors were within 10 nm, similar to the results with the hybrid sampling method and 0.004 s inverse feeding time, maintaining excellent surface topography fidelity to the designed geometry across the whole concave lenslet array.

## 4. Conclusions

In this paper, using real-time monitoring slow-tool servo position following error, different fabrication parameters were quantitatively studied and optimized, uniform and high-fidelity surface topography across the actual whole lenslet array was achieved and validated by actual fabricated results. Specifically, through PMAC real-time monitoring position following error at the lenslet array outer region, this paper quantitatively studied the impact of different sampling strategies, inverse time feed and arc-length, along with their associated total machining time consumption for STS. It provides a quick, effective, and detailed reference for both convex and concave lenslet array cutting parameter selection. In the end, a zonal machining strategy was demonstrated, and uniform and high-fidelity surface topography across the whole lenslet array was also achieved.

## Figures and Tables

**Figure 1 micromachines-13-01564-f001:**
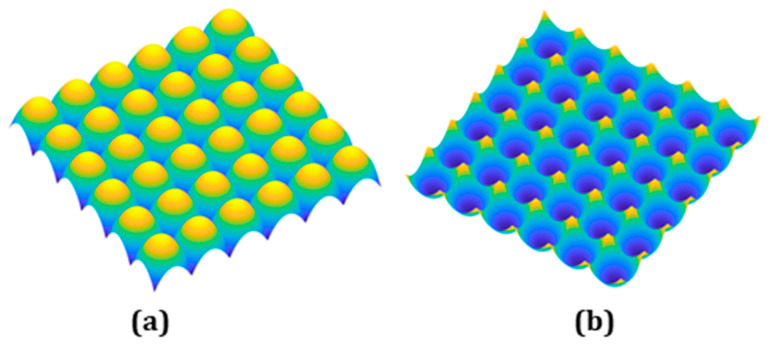
The 3D models of (**a**) convex and (**b**) concave lens arrays.

**Figure 2 micromachines-13-01564-f002:**
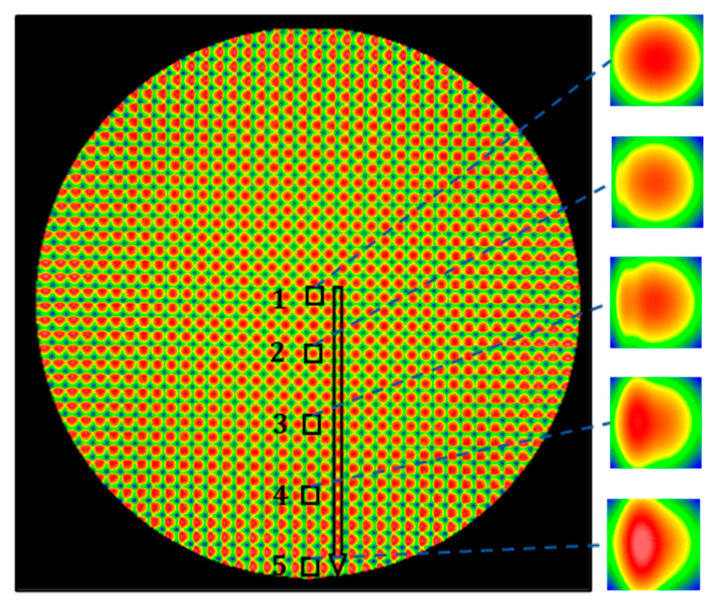
Lenslet array fabricated with the constant-angle strategy, with a constant-angle of 0.25 degrees and an inverse time feed of 0.001 s. The surface is measured and stitched together with a Zygo Newview 8300 optical surface profilometer. Five lenslets were selected for further analysis.

**Figure 3 micromachines-13-01564-f003:**
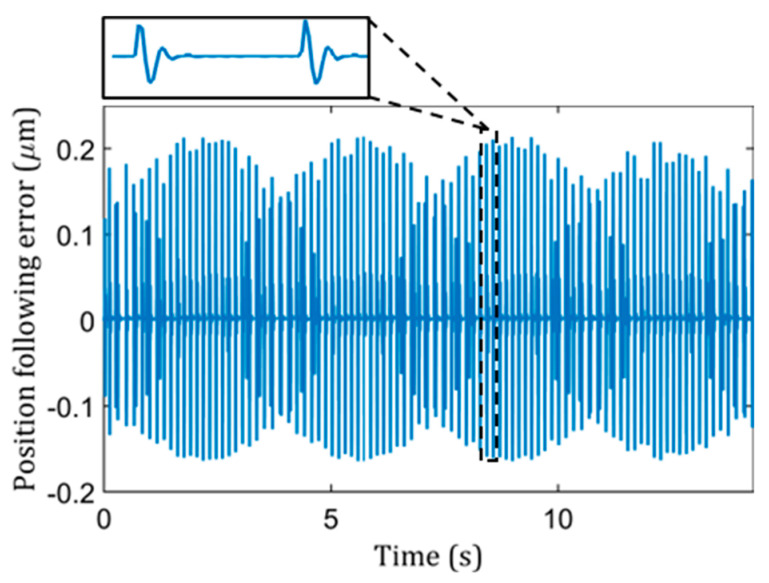
PMAC data acquired for position following error for the z-axis.

**Figure 4 micromachines-13-01564-f004:**
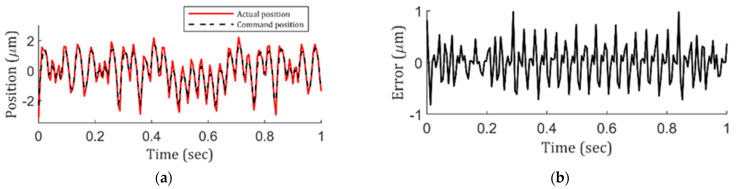
(**a**) Actual vs. command position in the z-axis of the STS with 0.001 s inverse time feed and 0.25° constant-angle cutting condition and corresponding (**b**) position following error.

**Figure 5 micromachines-13-01564-f005:**
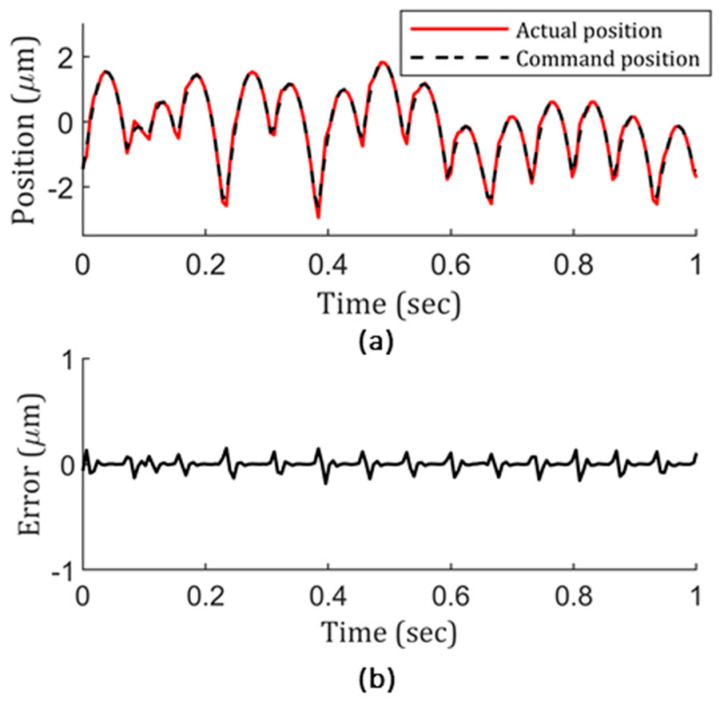
(**a**) Actual vs. command position in the z-axis of STS with a 0.004 s inverse time feed and a 0.25° constant-angle cutting condition and corresponding (**b**) position following error.

**Figure 6 micromachines-13-01564-f006:**
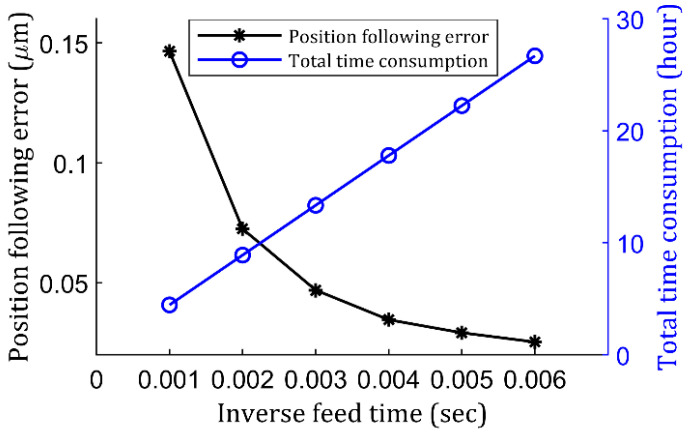
Position following error and total time cost verse inverse time feed study.

**Figure 7 micromachines-13-01564-f007:**
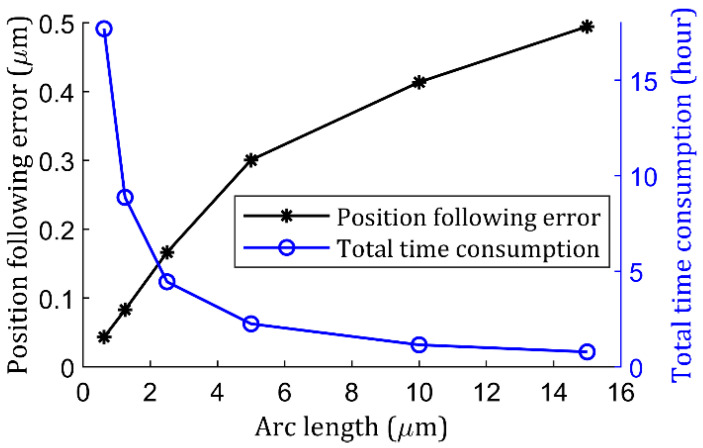
Position following error and total time consumption verse arc-length study.

**Figure 8 micromachines-13-01564-f008:**
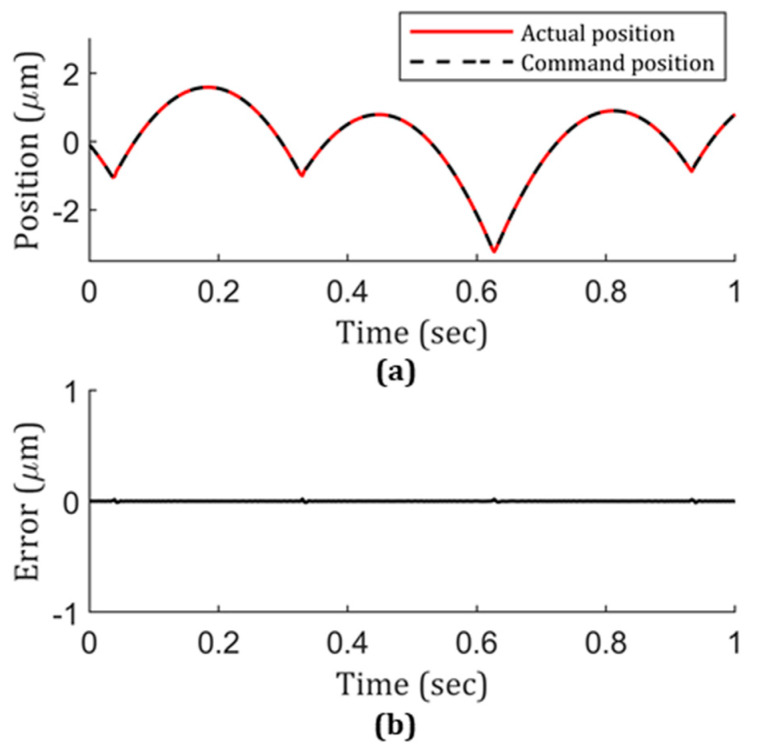
(**a**) Actual vs. command position on the z-axis of STS with a 0.004 s inverse time feed and a 2.5 µm constant-arc cutting condition and corresponding (**b**) position following error.

**Figure 9 micromachines-13-01564-f009:**
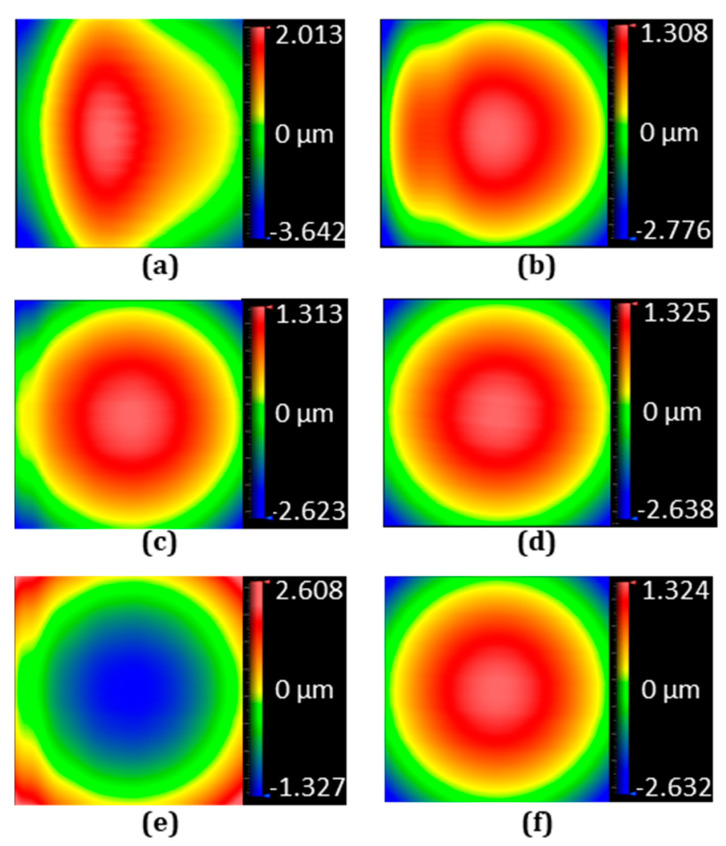
Measured surface shape of the 5th lenslet indicated in Figure 2 under different cutting parameters. Convex lenslet arrays with (**a**) 0.25° constant-angle sampling and 0.001 s inverse time feed, (**b**) 0.25° constant-angle sampling and 0.004 s inverse time feed, (**c**) 2.5 µm constant-arc sampling and 0.001 s inverse time feed, (**d**) 2.5 µm constant-arc sampling and 0.004 s inverse time feed, (**e**) concave lenslet array with 2.5 µm constant-arc sampling and 0.001 s inverse time feed, and (**f**) convex lens array with 2.5 µm constant-arc sampling in the outer region and 0.25° constant-angle sampling in the center region and 0.004 s inverse time feed. The measurement is performed with a Zygo Newview 8300 optical surface profilometer.

**Figure 10 micromachines-13-01564-f010:**
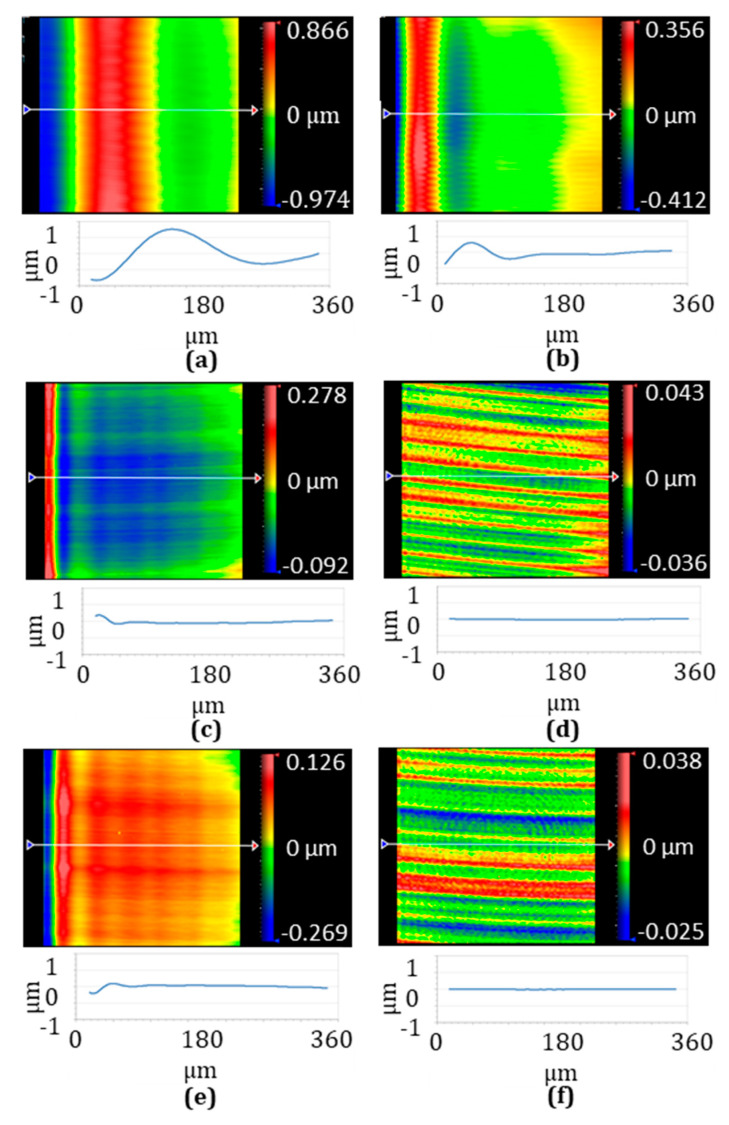
Surface error after removing the designed spherical shape for the lenslet in Figure 9. Similarly, convex lenslet arrays with (**a**) 0.25° constant-angle sampling and 0.001 s inverse time feed, (**b**) 0.25° constant-angle sampling and 0.004 s inverse time feed, (**c**) 2.5 µm constant-arc sampling and 0.001 s inverse time feed, (**d**) 2.5 µm constant-arc sampling and 0.004 s inverse time feed, (**e**) concave lenslet array with 2.5 µm constant-arc sampling and 0.001 s inverse time feed, and (**f**) convex lens array with 2.5 µm constant-arc sampling in the outer region and 0.25° constant-angle sampling in the center region and 0.004 s inverse time feed.

**Figure 11 micromachines-13-01564-f011:**
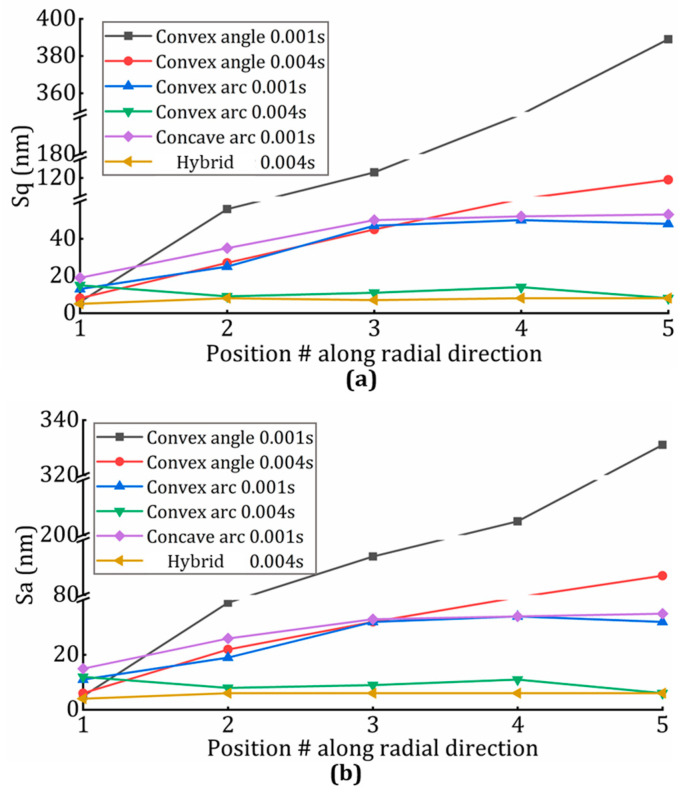
Sq and Sa of lenslets at different positions indicated in Figure 2 for different cutting parameters. Sq and Sa are shown in (**a**,**b**) after removing the designed spherical surface.

**Figure 12 micromachines-13-01564-f012:**
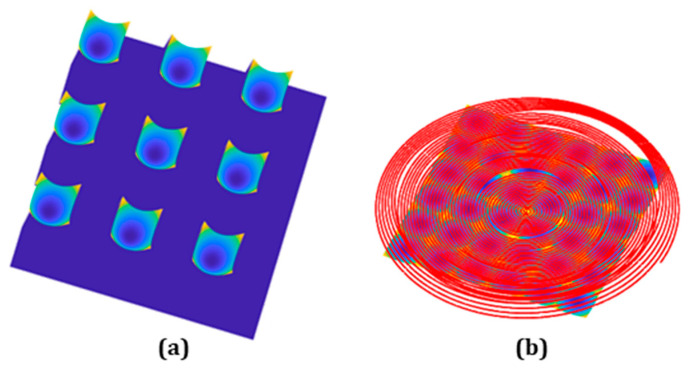
Tool path for fabricating lenslet array with concave lenslet. (**a**) One of the four lenslets is selected to fit the NURBS for the tool path in (**b**).

**Figure 13 micromachines-13-01564-f013:**
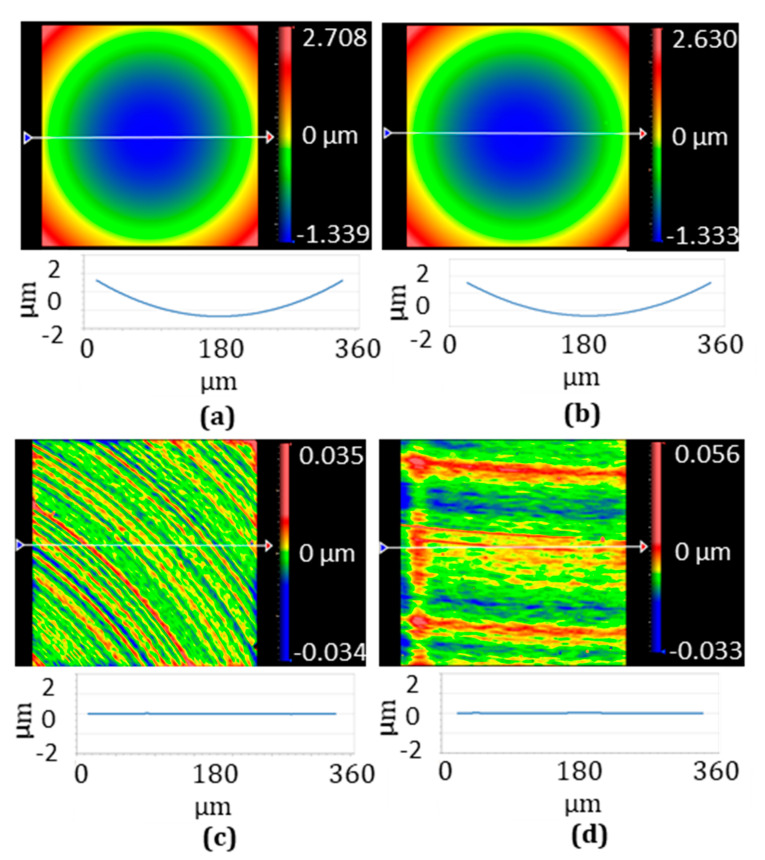
Measured surface of the concave lenslets fabricated with the tool path in Figure 12b. (**a**,**c**) Measured surface shapes of the concave lenslet in the center region before and after removing the designed spherical surface. (**b**,**d**) The results for the concave lenslet in the outer region.

## Data Availability

All presented in the paper, details are available upon request.

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
