# Peer review of "Experimental Studies on Fabricating Lenslet Array with Slow Tool Servo"

_micromachines, 2022, doi:10.3390/mi13101564_

Round 1

Reviewer 1 Report

This manuscript deals with parameter optimization studies on lenslet array fabrication with slow tool servo. The authors applied sampling strategy, inverse time feed, and others. Validation was done by surface morphology test, which supports the effectiveness of this method for lenslet arrays with convex and concave geometry. This manuscript is well-written and concise. I have only a few comments for improving the quality of this manuscript.

- I believe the effectiveness of the proposed method for improving the quality of lenslets. On the other hand, the reviewer is wondering about the scientific/engineering soundness. Could you please highlight the main claim of your work? 

- Recently, many different types of lens arrays have been developed for various applications. The authors could add some references, such as multi-focal lens arrays (e.g., Micromachines 11, 1068 (2020))

- The authors could comment on the scaling issues of the proposed techniques. 

Reviewer 2 Report

This is a well performed study on the accuracy of lenslet array fabrication. The work is clearly presented and I would only encourage the authors to make minor English edits, by proof reading and making sure that singular and plural are used correctly, as at present this distracts a bit from the reading.
